# Huangshan Maofeng Green Tea Extracts Prevent Obesity-Associated Metabolic Disorders by Maintaining Homeostasis of Gut Microbiota and Hepatic Lipid Classes in Leptin Receptor Knockout Rats

**DOI:** 10.3390/foods11192939

**Published:** 2022-09-20

**Authors:** Guohuo Wu, Wei Gu, Huijun Cheng, Huimin Guo, Daxiang Li, Zhongwen Xie

**Affiliations:** 1State Key Laboratory of Tea Plant Biology and Utilization, School of Tea and Food Sciences and Technology, Anhui Agricultural University, Hefei 230036, China; 2School of Life Sciences, Anhui Agricultural University, Hefei 230036, China; 3Center for Biotechnology, Anhui Agricultural University, Hefei 230036, China

**Keywords:** obesity, Huangshan Maofeng green tea, *Lepr*^−/−^ rats, gut microbiota, lipidomics

## Abstract

Huangshan Maofeng green tea (HMGT) is one of the most well-known green teas consumed for a thousand years in China. Research has demonstrated that consumption of green tea effectively improves metabolic disorders. However, the underlying mechanisms of obesity prevention are still not well understood. This study investigated the preventive effect and mechanism of long-term intervention of Huangshan Maofeng green tea water extract (HTE) on obesity-associated metabolic disorders in leptin receptor knockout (*Lepr*^−/−^) rats by using gut microbiota and hepatic lipidomics data. The *Lepr*^−/−^ rats were administered with 700 mg/kg HTE for 24 weeks. Our results showed that HTE supplementation remarkably reduced excessive fat accumulation, as well as ameliorated hyperlipidemia and hepatic steatosis in *Lepr*^−/−^ rats. In addition, HTE increased gut microbiota diversity and restored the relative abundance of the microbiota responsible for producing short chain fatty acids, including *Ruminococcaceae*, *Faecalibaculum*, *Veillonellaceae*, etc. Hepatic lipidomics analysis found that HTE significantly recovered glycerolipid and glycerophospholipid classes in the liver of *Lepr*^−/−^ rats. Furthermore, nineteen lipid species, mainly from phosphatidylcholines (PCs), phosphatidylethanolamines (PEs), and triglycerides (TGs), were significantly restored increases, while nine lipid species from TGs and diglycerides (DGs) were remarkably recovered decreases by HTE in the liver of *Lepr*^−/−^ rats. Our results indicated that prevention of obesity complication by HTE may be possible through maintaining homeostasis of gut microbiota and certain hepatic lipid classes.

## 1. Introduction

Obesity is becoming an important public health issue around the world, and it is a cluster of the dangerous risk factors linked to metabolic syndrome, type 2 diabetes, hypertension, and cardiovascular disease [1,2,3]. Recent studies have shown that obese individuals are more likely to develop COVID–19 [4,5]. According to the definition of obesity by the World Health Organization (WHO), the ranges of body mass index (BMI, in kg/m^2^) ≥ 25 are overweight, and BMI ≥ 30 are obesity. Right now, 1.9 billion adults are overweight, over 650 million of them are obese, and the prevalence of obesity is expected to rise in the future [6,7]. Apparently, there is a desperate need for effective management of obesity. However, treating obesity with medicines is commonly associated with a variety of side effects [8,9,10], such as gastrointestinal dysfunction, hepatotoxicity, statin–related myotoxicity, etc. Interestingly, studies have shown that foods and their functional ingredients can prevent obesity and other metabolic diseases with fewer side effects [11,12,13]. Thus, the investigation of dietary interventions in the prevention of obesity and their potential mechanisms is of important significance to public health. The development of obesity may be a result of a combination of genetic and environmental factors [7,14], such as unhealthy eating habits and lifestyle. Recent researches have shown that the composition and metabolic functions of gut microbiota play a crucial role in obesity development [15,16]. Emerging evidence has revealed that foods and their functional components can affect the metabolism of lipids and prevent obesity through altering gut microbiota composition [17,18,19], which indicates that gut microbiota may serve as a potential new target for obesity prevention.

Tea is one of the most popular drinks in the world, and contains a wide variety of bioactive compounds [20], such as catechins, theanine, caffeine, polysaccharides, etc. Previous research has indicated that drinking tea or consuming tea ingredients has health benefits to the public, such as reducing body weight, alleviating metabolic syndrome, preventing cardiovascular diseases and cancer, and improving cognitive ability [17,21,22,23]. Recent studies reported that consuming tea could modulate gut microbiota to improve host metabolism, indicating a possible role of microbiota in regulating health benefits related to tea drinking, such as hypoglycemia and weight loss [23,24,25]. Liu et al. reported that Fuzhuan brick tea attenuates obesity by altering gut bacteria in high-fat diet-induced mice [26]. Chen et al. also reported that tea polyphenols lower blood glucose levels by changing gut microbiota in db/db mice [27]. Our previous studies found that large yellow tea ameliorated metabolic disorders by reducing lipogenesis and altering microbiota in leptin receptor knockout rats [28]. Huangshan Maofeng green tea (HMGT) is one of the most famous green teas, which has been produced and consumed for a thousand years in China [29]. The tea trees used for making HMGT grow at altitudes above 700 m around Huangshan Mountain in Anhui province, China [30]. Cui et al. reported that HMGT extract protected mice from CCl4–induced liver damage [31]. Our previous studies proved that HMGT supplementation prevented hypertension induced by desoxycorticosterone acetate and salt in old C57BL/6 mice [32]. However, the effect and mechanism of HSMF in preventing obesity complication are not well understood. Notably, potential roles of gut microbiota and lipidomic classes regulated by HMGT in obesity prevention remains unclear.

The liver is an important organ that regulates nutrition metabolism along with energy balance [33]. Dysregulation of hepatic lipid metabolism can result in hepatic steatosis, obesity, and other comorbidities [28,34,35]. However, traditional biochemical data cannot provide detailed information on which classes of lipids or lipidomic pathways are compromised with obesity complication. Lipidomics has recently emerged as a new way of measuring lipid classes and identifying biomarkers associated with lipid metabolism [36]. Shon et al. studied the obesity-prevention effect of *Chrysanthemum morifolium* Ramat leaves using plasma lipidomics [37]. In addition, Nam et al. reported that green tea ethanol extracts improved lipid profiles in mice fed a high-fat diet using lipidomics analysis [38]. So far, few studies have utilized lipidomics to explore how green tea alters the profiles of hepatic lipid metabolism.

The *Lepr*^−/−^ rat is a newly developed leptin receptor knockout rat model characterized by obesity, glucose intolerance, and dyslipidemia, which is proven to be a suitable animal model to study obesity complication [28,39,40]. In this study, the combination of both gut microbiota and hepatic lipid profiles data were applied to investigate the preventive mechanism of obesity-associated metabolic disorders by long term intervention of Huangshan Maofeng green tea water extracts (HTE) in *Lepr*^−/−^ rats model.

## 2. Materials and Methods

### 2.1. Chemical Analysis of Huangshan Maofeng Green Tea Water Extracts

HSMF samples were obtained from the Huangshan Maofeng Tea Group Company (Anhui, China). HSMF was powdered and extracted three times by boiled water with ultrasound assistance according to our previous method [12]. Afterwards, the extractions were concentrated under vacuum and lyophilized, yielding HSMF water extracts (HTE) which were stored at minus 80 °C. Table 1 lists the main ingredients in HTE determined by our previous method [28].

### 2.2. Animals Experiment

Leptin receptor knockout (*Lepr*^−/−^) rats (Sprague-Dawley background) were obtained from the Chinese Academy of Medical Sciences (Beijing, China). The rats were free to drink and eat and were kept in the specific pathogen free laboratory animal center (temperature 22 ± 1 °C, relative humidity 50 ± 5% with 12 h light−dark cycle). At the age of seven weeks, *Lepr*^−/−^ rats were randomly divided into the *Lepr*^−/−^ control group (KO) (*n* = 6) and the *Lepr*^−/−^ with HTE intervention group (KH) (*n* = 6). Wild type littermates are considered as the WT control group (WT) (*n* = 6). The rats in the KH group were given 700 mg/kg HTE by oral gavage once every day for a total of 24 weeks, and the rats in the control group were administered the same amount of water. The HTE was prepared freshly every day at the concentration of 100 mg/mL just before administration. The ethical protocol for animal study was approved on 4 April 2019 by the Institutional Animal Care and Use Committee of the Anhui Agricultural University (ethical code: AHAU 2019–034).

The TP23512 diet for rats was purchased from Trophic Animal Feed Co., Ltd. (Nantong, China, Appendix A). Fat mass and lean mass were determined as body composition analysis by using minispec LF90II Analyzer (Bruker, Germany) according to our previous method [41]. At the end of the experiment, rats were sacrificed after being anesthetized with pentobarbital sodium following a fast of 12 h. The blood, liver, inguinal, and epididymal adipose tissues were collected for further study. The formula for calculating Lee’s index is as follows: Lee’s index = body weight (g) ^ (1/3) × 1000/body length (cm) [42].

### 2.3. Serum Biochemical Analysis

The levels of triglyceride (TG), total cholesterol (T-CHO), low–density lipoprotein cholesterol (LDL-C) in serum and liver tissue, as well as serum aspartate transaminase (AST) and alanine aminotransferase (ALT) levels were determined using activity assay kits (Jiancheng Biotech, Nanjing, China). In addition, the levels of free fatty acid (FFA) were determined using enzyme linked immunosorbent assay (ELISA) kits (Jianglai Biotech, Shanghai, China) according to operating instructions.

### 2.4. Histopathological Analysis

Histopathological analysis was carried out according to the previous method [43]. Briefly, formalin-fixed liver and adipose tissue samples were paraffin-embedded, sectioned at 5 μm, and stained with hematoxylin and eosin (H & E) for histological analysis. In addition, formalin-fixed liver tissue samples were sectioned at 10 μm and stained with Oil red O (ORO) staining. Images were taken by microscope at 100× or 400× magnification. The area of the white adipocytes and the hepatic lipid were determined using Image-J software (NIH, Bethesda, MD, USA).

### 2.5. RNA Isolation and Real-Time PCR Analysis

Real-time PCR analysis was performed according to our previous method [44]. To measure the expression of hepatic lipogenic gene in the liver samples, total RNA was extracted by RNA-easy Isolation Reagent (Vazyme, Nanjing, China). cDNA was synthesized using HiScript II cDNA Synthesis kit (Vazyme, Nanjing, China). Quantitative real-time PCR (qRT-PCR) was conducted using AceQ qPCR SYBR Green kit (Vazyme, Nanjing, China). The relative quantification was performed using the 2^−ΔΔCt^ method. The primer sequences are listed in Appendix A.

### 2.6. Analysis of Liver Lipidomics

Each sample preparation used 30 mg of liver tissue. The tissue sample was pulverized in liquid nitrogen. Then, 300 μL of PBS was added to liver tissue powder and vortexed to mix well. Next, 429 μL of methyl tert–butyl ether, 342 μL of methanol, and 429 μL of H_2_O was added and vortexed for 1 min, followed by centrifugation at 2000× *g* at 4 °C for 5 min. After that, the supernatant was evaporated at −20 °C, and then 100 μL isopropanol was added to the dried mixture, which was vortexed for 3 min. The mixture samples were centrifuged again at 12,000× *g* for 15 min. The supernatant was collected in a glass vial for analysis.

Lipidomic analysis was accomplished using Ultimate HPLC system (ThermoFisher Scientific, Waltham, MA, USA) equipped with XBridge BEH HILIC (100 mm × 2.1 mm, 2.5 μm; Waters, Milford, MA, USA). MS detection was executed on an Q Exactive Focus Oribitrap MS/MS system (Thermo Fisher Scientific, Waltham, MA, USA). Mass spectra and chromatograms were acquired and processed with Xcalibur. The Automated Identification Engine for Lipidomics Version 4.0 software was used to determine the chemical formula for product and precursor ions. Univariate and multivariate analysis were conducted by metaboanalyst 5.0. The data analysis for the study was conducted according to previous methods [45].

### 2.7. Analysis of Gut Microbiota

Colon content samples were collected and frozen in liquid nitrogen immediately after the rats were sacrificed, and then stored at −80 °C until use. Total DNA in colon content samples was isolated, amplified, and constructed using SMRT Bell library according to our previous methods [28]. The colonic microbiota composition was assessed by the PacBio RS II platform and QIIME package was used for obtaining high–quality sequences. For precise operational taxonomic unit (OTU) analysis, the high–quality sequences with similarity > 97% were classified into OTU through UCLUST. Based upon an 80% confidence threshold, the OTU representative sequences were taxonomized using the Ribosomal Database Project II database.

### 2.8. Quantification of Fecal Short–Chain Fatty Acids

The samples of rat feces were collected and frozen in liquid nitrogen, and then stored at −80 °C until analysis was performed. The level of short–chain fatty acids (SCFAs) in the feces was extracted and measured using a gas chromatography method according to a previous study [46]. Determination of fecal SCFAs was performed by gas chromatography equipped with a flame ionization detector (Agilent 5977, Santa Clara, CA, USA). The concentrations of fecal SCFAs were quantified using external calibration curves. 

### 2.9. Statistical Analysis

Results are presented as mean ± SEM. Student’s *t*–test was used to assess the significance of the statistical difference between the two groups, and multiple group comparisons were performed by one–way ANOVA. Using GraphPad Prism, we first ascertained whether data showed normal distribution. Then, equal normality test was performed. Parametric analysis was performed using one-way ANOVA followed by Tukey’s test. This reaches statistical significance when the *p*-value was less than 0.05.

## 3. Results

### 3.1. Main Chemical Compounds in Huangshan Maofeng Green Tea Water Extract

The major catechins, polysaccharide, caffeine, and amino acid in HTE were determined and shown in Table 1. According to the results, total catechins (381.84 ± 4.56 mg/g), polysaccharide (153.89 ± 1.36 mg/g), and caffeine (106.92 ± 0.93 mg/g) are the main chemical components in HTE. Large yellow tea is one of the most famous traditional yellow teas in China, which is known to prevent metabolic syndrome significantly [12,28]. Comparing the main chemical compounds in large yellow tea extracts (LTE), the HTE contained a higher concentration of total catechins, total amino acid, and (−)-epigallocatechin gallate (EGCG) (*p* < 0.05). However, the content of (−)-gallocatechin gallate (GCG), (−)-gallocatechin gallate (GC), polysaccharide, and caffeine in HTE was evidently lower than that of LTE (*p* < 0.05).

### 3.2. HTE Attenuated Phenotypes of Obesity in Lepr^−/−^ Rats

The *Lepr*^−/−^ rat is characterized by obesity and dyslipidemia. To evaluate the effect of HTE on obesity in *Lepr*^−/−^ rats, we administered 700 mg HTE per kg body weight to rats for 24 weeks. As shown in Figure 1A, the *Lepr*^−/−^ rats showed rapid weight gain compared to WT rats. After 6 weeks of intervention, the HTE group showed significantly lower body weights compared to the *Lepr*^−/−^ control group (*p* < 0.05), and this preventive effect persisted until to the end of 24-week intervention. An analysis of body composition revealed a significant reduction of fat mass gain and an increase of lean mass loss in *Lepr*^−/−^ rats following HTE intervention (Figure 1B,C). Lee’s index is an effective parameter in evaluating the obesity degree of rats. Compared to WT rats, the Lee’s index of *Lepr*^−/−^ rats showed a significant increase. However, HTE supplementation obviously decreased the Lee’s index of *Lepr*^−/−^ rats (Figure 1D). In addition, weights of liver and adipose tissues, including inguinal and epididymal fat, were remarkably heavier in the *Lepr*^−/−^ group than that in the WT group. The weights of liver and adipose tissues were lower in the KH group when compared with the KO group (*p* < 0.05) (Figure 1E–G and Appendix A). H & E staining results showed that the sizes of the inguinal and epididymal adipocytes were reduced in the KH group as compared to the KO group (*p* < 0.05) (Figure 1H–K). We measured food intake of each group of rats to determine if HTE affected energy intake in *Lepr*^−/−^ rats. The results indicated that there were no significant differences in food intake between the *Lepr*^−/−^ and KH groups (Appendix A). However, the food intake in *Lepr*^−/−^ rats was elevated significantly by 55.29% in comparison with WT rats (*p* < 0.001). 

### 3.3. HTE Improved Dyslipidemia in Lepr^−/−^ Rats

The measurement results for serum biochemical parameters including TG, T-CHO, LDL-C, HDL-C, and FFA levels are presented in Figure 2. The results showed that serum TG, TCHO, LDL-C, HDL-C, and FFA levels in the KO group showed all significant increase when compared with those in the WT group (*p* < 0.001). Supplementation with HTE significantly decreased serum TG, TC, LDL-C, and FFA levels (*p* < 0.05), although the levels of serum HDL-C did not appear to be altered by supplementation of HTE. The results suggested that HTE could improve dyslipidemia in *Lepr*^−/−^ rats.

### 3.4. HTE Ameliorated Hepatic Steatosis in Lepr^−/−^ Rats

Obesity is frequently associated with hepatic steatosis. As shown in Figure 3A, H & E staining showed that liver tissue from the WT group exhibited larger polygonal cells with clear round nuclei and less lipid accumulation, whereas liver tissue from the *Lepr*^−/−^ group showed diffuse hepatic fatty infiltration. However, the hepatic steatosis in *Lepr*^−/−^ rats was dramatically reduced after HTE intervention (the histopathology of the livers with lower magnification; see Appendix A). Oil red staining showed that the accumulation of higher levels of lipids in the liver of *Lepr*^−/−^ rats was significantly reduced after administration of HTE (Figure 3B,C). The serum ALT and AST activities, used as the biochemical markers of hepatic injury, are shown in Figure 3D,E, respectively. The KO group showed remarkably elevated serum ALT and AST activities than the WT group (*p* < 0.001). However, HTE supplementation obviously reduced serum ALT and AST activities by 55.77% and 35.64% (*p* < 0.05), respectively. Moreover, the levels of liver TG, T-CHO, and LDL-C in the KH group were reduced when compared with those in the KO group (*p* < 0.05) (Figure 3F–H). Results all indicated that HTE alleviated hepatic steatosis in *Lepr*^−/−^ rats.

### 3.5. HTE Reduced the Expressions of Lipogenesis Related Genes in Liver of Lepr^−/−^ Rats

Previous studies have shown that dysregulation of hepatic lipogenesis genes is associated with obesity [47]. Therefore, the expression levels of lipogenesis related genes were evaluated to investigate the regulating effect of HTE on lipid metabolism in *Lepr*^−/−^ rats (Figure 4). Our results showed that the expression levels of *Srebf1*, *Thrsp*, *Pparγ*, *Accα*, *Fasn*, *Scd1*, and *Hmgcr* remarkably increased in the KO group compared to the WT group. However, HTE supplementation significantly reduced the expression levels of *Lxrα*, *Srebf1*, *Thrsp*, *Pparγ*, *Accα*, *Fasn*, *Scd1*, and *Hmgcr* (*p* < 0.05), which is consistent with lower levels of serum FFA, TG, and TC in the KH group. 

### 3.6. HTE Improved Hepatic Lipid Profiles by Lipidomics Analysis

The liver is an important organ for nutrient metabolism and lipogenesis. To understand the preventing obesity effects of HTE in *Lepr*^−/−^ rats, we further conducted hepatic lipidomics to analyze the liver lipid profiles. The typical total ion current chromatograms of liver samples were analyzed with the UHPLC-Orbitrap-MS/MS system, which yielded 675 metabolite ion features (Appendix A). Based on the lipid metabolites, a principal component analysis (PCA) was performed to visualize the data. Results showed that the lipid profiles from liver of the WT, KO, and KH group were clearly separated (Figure 5A), indicating a significant difference in lipid profiles between the three group samples. We then used partial least squares discrimination analysis (PLS-DA) to analyze potential discrimination among sample groups. The result also revealed that the three group samples were clearly separated from each other (Figure 5B). Furthermore, the permutation tests for PLS-DA model validation resulted in an R^2^ and Q^2^ of 0.645 and –0.35, which indicated that the models had been established successfully without being overfit (Figure 5C). The plot of volcanoes was used to identify lipid compounds responsible for discrepancies. Using the standard of FC ≥ 2 or ≤ 0.5 and *p* ≤ 0.05, a total of 306 significantly different lipid species were identified for the KO group versus the WT group (Figure 5D). Moreover, 59 significantly different lipid species were identified when the KH group and the KO group were compared (Figure 5E). 

A heatmap was generated to show the changes of the lipid classes in the liver of rats. The results showed that the level of diglycerides (DGs), triglycerides (TGs), and phosphatidylethanol (PEts) remarkably increased in the KO group than the WT group. The following lipid classes, such as phosphatidylcholines (PCs), lyso-phosphatidylcholines (LPCs), phosphatidylethanolamines (PEs), lyso-phosphatidylethanolamines (LPEs), dimethyl-phosphatidylethanolamines (dMePEs), phosphatidylserines (PSs), phosphatidylglycerols (PGs), phosphatidylinositols (PIs), cardiolipins (CLs) in glycerophospholipids, and sphingomyelins (SMs) in sphingolipids were dramatically decreased in the KO group when compared to the WT group (Figure 5F). However, after HTE intervention, the lipid classes of *Lepr*^−/−^ rats were reversed to the level of WT rats. The lipid metabolites significantly affected by HTE are shown in Figure 6 and Figure 7. The results showed that the 19 lipid classes, including 14 glycerophospholipid classes (PC (36: 2), PC (36: 4), PC (37: 2), PC (37: 5), PC (42: 7), PE (34: 2), PE (36: 2), PE (36: 3), PE (36: 4), PE (40: 5), PS (36: 4), PS (38: 4), Pet (33: 2), CL (20: 5/18: 2/18: 2/18: 2)) and 5 glycerolipids (TG (16: 1/18: 2/20: 4), TG (16: 0/18: 3/22: 6), TG (18: 3/18: 2/22: 6), TG (18: 3/18: 3/22: 6), TG (18: 3/20: 5/22: 6)), were significantly increased, while the 9 lipid classes, including 1 glycerophospholipid class (PC(38: 5)) and 8 glycerolipids (DG (16: 0/18: 1), DG (18: 1/18: 1), TG (18: 0/18: 0/18: 1), TG (18: 0/18: 1/18: 1), TG (16: 0/18: 1/20: 3), TG (21: 1/18: 1/18: 1), TG (18: 0/18: 0/20: 4), TG (18: 1/18: 1/22: 3)), were significantly decreased in *Lepr*^−/−^ rats after HTE supplementation. Furthermore, according to the correlation analysis, the 28 lipid species were significantly correlated with the phenotypes of obesity, including serum TG, serum TC, serum LDL-C, liver TG, liver TC, liver LDL-C, etc. Additionally, 28 differential metabolites were identified among the three groups based on *p* values. Among these 28 lipid species, the level of PE (36:3), PS (36:4), CL (20:5/18:2/18:2/18:2), and TG (18:3/18:2/22:6) were strongly negatively correlated with obesity related disorder, whereas the level of DG (16:0/18:1) were strongly positively correlated with obesity related indexes (Figure 8).

### 3.7. HTE Increased Gut Microbiota Diversity in Lepr^−/−^ Rats

It has been shown that diet mediating biological function is closely related to gut microbiota modulation [48]. Therefore, we examined the effects of HTE on gut microbiota homeostasis using 16S rRNA sequencing. A total of 844 bacterial OTUs were obtained, and 608 OTUs were found in the WT group, whereas 244 OTUs were identified in the *Lepr*^−/−^ group. After administration with HTE, the number of OTUs in the KH group increased to 443 (Figure 9A). At phylum level, the abundance of *Firmicutes* was significantly increased in the KO group, while the abundance of *Proteobacteria*, *Bacteroidetes*, *Deferribacteres*, *Actinobacteria*, *Fusobacteria*, and *Melainabacteria* were remarkably reduced in the KO group as compared to the WT group (Figure 9B). After administration with HTE, the microbiota composition at the phylum level in the KH group was revised to the WT group (Figure 9B). As shown in Figure 9C, by comparing the *Lepr*^−/−^ group to the WT group, we found that the relative abundance of *Bacteroidetes* was decreased significantly, while *Firmicutes* were increased, which resulted in a higher F/B ratio in the KO group. After HTE intervention, the F/B ratio in the KO group was dramatically reduced. We further analyzed the microbiota composition at the genus level and found that the richness of gut microbiota in the KO group was reduced when compared to the WT group and KH group (Figure 9D). To investigate the effect of HTE on α–diversity of gut microbiota in *Lepr*^−/−^ rats, including the observed species index, the Chao 1 index, and the Shannon and Simpson indexes, we found that the KH group showed significant increases in these indexes when compared with the KO group (Figure 9E–H). According to the principal coordinates analysis (PCoA) based on the unweighted and weighted UniFrac, three groups of rats had significantly different gut microbiota compositions and structures (Figure 9I,J).

### 3.8. HTE Altered the Gut Microbiota Composition in Lepr^−/−^ Rats

First, we analyzed the abundance of top 30 genera in gut microbiota among different groups of rats, and the results revealed that the microbiota composition of the *Lepr*^−/−^ group was significantly different from that of the WT group (Figure 10A). The microbiota producing SCFAs, including *Odoribacter*, *Blautia*, *Bacteroides*, *Intestinimonas*, *Faecalibaculum*, unidentified_*Ruminococcaceae*, unidentified_*Veillonellaceae*, *Turicibacter*, and *Parabacteroides*, decreased and opportunistic pathogens (*Streptococcus*) increased in the *Lepr*^−/−^ group compared to the WT group, which indicated that gut microbiota dysbiosis had occurred in the *Lepr*^−/−^ group. Second, we ranked the microbiota according to the *p* values by comparing the *Lepr*^−/−^ group with the KH group. At the genus level, compared with the WT group, the relative abundance of *Lactobacillus* was significantly higher, while the relative abundances of unidentified_*Ruminococcaceae*, *Faecalibaculum*, and *Mucispirillum* were dramatically reduced in the *Lepr*^−/−^ group (*p* < 0.05) (Figure 10B). However, we found that HTE supplementation significantly reversed this balance to the WT group in the KH group (Figure 10B). In addition, a spearman correlation analysis was conducted to investigate the relationship between top 16 genera of gut microbiota and obesity complication-related parameters (Figure 10C). The correlation analysis showed that the abundance of *Faecalibaculum*, *Intestinimonas*, unidentified_*Ruminococcaceae*, *Ileibacterium*, unidentified_*Enterobacteriaceae Blautia*, *Lachnoclostridium*, and *Allobaculum* were significantly negatively correlated with obesity parameters, whereas the abundance of *Lactobacillus*, *Turicibacter*, and *Candidatus_Saccharimonas* were positively associated with obesity related indexes. Additionally, the abundance of unidentified_*Lachnospiraceae* and *Mucispirillum* was negatively associated with the serum TG, serum LDL-C, the AUC of OGTT, and leptin levels. The abundance of *Parabacteroides* was negatively associated with the HOMA-IR index. These results showed that HTE supplementation resulted in alterations in gut microbiota composition that were closely related to improving obesity complication in *Lepr*^−/−^ rats. Furthermore, we determined the effects of HTE intervention on the SCFAs producing by gut microbiota in fecal of rats. The results showed that except for acetic acid, HTE supplementation significantly increased the levels of fecal propanoic acid, butyric acid, isobutyric acid, pentanoic acid, and isopentanoic acid in *Lepr*^−/−^ rats (Appendix A).

## 4. Discussion

Obesity complications are cluster disorders, including obese, dyslipidemia, diabetes, and fatty liver. There is an increasing body of evidence which suggests that dietary supplementation with tea and tea extracts may be a practical strategy for preventing or treating obesity complication in humans and animals [21]. The majority data from mice studies reported that green tea has health benefits on obesity prevention, and its polyphenols are the active ingredients believed to be responsible for these health benefits [49,50]. HMGT is one of the famous green teas produced and consumed for a long history in Anhui, China. However, the effect on obesity prevention of HMGT has not been investigated. The present study systematically investigated the obesity prevention effect of long-term supplement of water extracts of HMGT using *Lepr*^−/−^ rats model. Our chemical analysis data showed that the water extracts of HMGT (HTE) was characterized as a polyphenol-enriched fraction containing high concentration of EGCG (178.14 ± 2.37% mg/g) as the major catechin, followed by polysaccharide (153.89 ± 1.36% mg/g), caffeine (106.92 ± 0.93% mg/g), and theanine (30.51 ± 1.43 mg/g). EGCG is one of the richest bioactive polyphenol components in tea and has been reported to have fat-reduction and weight-loss effects. Huang et al. reported that EGCG alleviated obesity and fatty liver development by reducing lipid and bile acid absorption in mice [51]. Choi et al. reported that EGCG reduces obesity by regulating Beclin1-dependent autophagy in white adipose tissues of mice [52]. Furthermore, studies have revealed that tea polysaccharides can reduce the risk of type 2 diabetes, obesity, as well as obesity-related diseases by modulating gut microbiota and metabolism [25,53]. Additionally, caffeine has been shown to be one of the most effective bioactive constituents in tea for reducing fat deposits and weight loss [54,55]. Here, we speculated that EGCG, polysaccharides, and caffeine from HTE all can contribute to prevention obesity complication in *Lepr*^−/−^ rats. Studies have shown that the health benefits of whole tea extracts are usually superior to those of single component, which may be a result of the synergistic effects of the various functional components found in tea [12,56]. Therefore, the synergistic effects and mechanisms of combination of multiple functional components in tea against obesity complication need to be further investigated in the future. Many previous studies have demonstrated that green tea supplement did not significantly affect lipid metabolism in wild type rats [57,58,59]. Therefore, a WT rat with HTE intervention group was not included in present study. Our results demonstrated that dietary supplement of HTE (700 mg/kg BW) efficiently prevented obesity related disorders, including overweight, hyperlipidemia, hepatic steatosis, and excessive adipose tissue accumulation in *Lepr*^−/−^ rats. The dose used in the present study (700 mg/kg BW) is equivalent to 15 g daily tea consumption for a 70 kg human, and this amount is reasonable for regular tea consumers on a daily basis [60].

Dyslipidemia is a characteristic of obesity complication, which can result in insulin resistance of the liver and nerve endings, and damage to the islets [61]. Previous studies have reported that the green tea extract can obviously decrease body weight and levels of TC, TG, and LDL-C [58,62]. Our data further showed that HTE reversibly decreased the levels of serum TC, TG, LDL-C, and FFA, but did not reduce the levels of serum HDL-C in *Lepr*^−/−^ rats model. Our previous study reported that supplementation of large yellow tea did not alter the concentration of HDL-C in db/db mice [12]. A recent paper found that instant dark tea induced the level of serum HDL-C in high-fat diet-fed rat [63]. The possible reason for difference between this result and our present data might be due to the different chemical profiles between HTE and instant dark tea. 

The liver is a vital organ for the regulation of lipid metabolism. An excessive accumulation of lipid in the liver is a characteristic of obesity, which can result in liver injury, fibrosis, and cirrhosis. In our study, as compared to the *Lepr*^−/−^ rat group, supplementation of HTE reduced hepatic lipid accumulation, and suppressed ALT and AST activities, alleviating liver injury in *Lepr*^−/−^ rats. It has been reported that tea and its functional ingredients can lower lipid accumulation by suppressing lipogenesis and promoting catabolism of lipids in the liver of rodent model [21]. Thus, we measured the expression levels of lipogenesis related genes by real-time PCR. The results indicated that the mRNA expression levels of *Lxr**α*, *Srebf1*, *Thrsp*, *Pparγ*, *Accα*, *Fasn*, *Scd1*, and *Hmgcr* were remarkably suppressed in the KH group when compared to the *Lepr*^−/−^ rat group. SREBP1 is a transcription factor that contributes to the differentiation of adipocytes and lipogenesis by regulating lipogenesis related genes, such as *Thrsp*, *Accα*, *Fasn*, and *Scd1* [64,65]. LXR*α* is a transcription factor that is critical for regulating hepatic lipid metabolism, which could significantly increase SREBP-1c expression directly, thereby promoting *de novo* lipogenesis in the liver [66,67]. *Thrsp* was demonstrated to be a lipogenic gene whose expression is controlled by LXRα and SREBP-1c [68,69]. In addition, PPARγ is a nuclear receptor regulating fatty acid synthesis and oxidation. Studies suggested that lipid catabolism was stimulated, and fatty acid synthesis was suppressed by inhibiting PPARγ expression or activity in the liver [47,70,71]. *Accα*, *Fasn*, *Scd1*, and *Hmgcr* are de novo lipogenesis related genes [72,73,74]. These results suggested that HTE could ameliorate fatty liver and protect the liver injury by suppressing the expression of wide-ranging lipogenesis genes in the liver of *Lepr*^−/−^ rats. 

This study further examined the lipid classes in the liver of *Lepr*^−/−^ rats after HTE supplementation using mass spectrometry-based lipidomic approaches [36]. We found that the relative abundance of glycerides in the liver of *Lepr*^−/−^ rats dramatically increases, whereas glycerophospholipids and sphingolipids significantly decrease when compared to WT rats. In a previous study, high fat diets resulted in an increase in liver glycerolipids (DGs and TGs), while a decrease in glycerophospholipids (PCs and PEs) and sphingolipids (SMs) resulted in abnormal lipid metabolism in mice [75]. Our results on leptin receptor knockout rats indicated that HTE supplementation induced obvious recovery of lipid classes similar to WT rats in liver of *Lepr*^−/−^ rats. 

Next, we screened the group specific lipid species. The 28 lipid species, including 13 glycerolipids and 15 glycerophospholipid classes, were significantly altered by HTE supplementation. Glycerolipids are the main lipids in the body as well as the major source of energy, including MGs, DGs, and TGs. Studies have shown that high DGs and TG levels in glycerolipids are indicative of fatty liver disease [76]. A high level of monounsaturated DGs in the liver is associated with insulin resistance and fatty liver [76,77]. In a study by Feng et al., the abundance of DGs in the livers of rats fed a high-fat diet increased, and the reduction of DGs was associated with the weight loss and lipid lowering effects of Tangeretin [78]. Our data indicated that the decreased level of hepatic DG (16:0/18:1) and DG (18:1/18:1) in *Lepr*^−/−^ rats might play an important role in the anti-obesity effect of HTE. Additionally, studies had shown that TG species with lower carbon numbers and double-bond content have been linked to increased cardiovascular disease and type II diabetes, while species with higher carbon numbers and higher double-bond content are linked to a decreased risk [79,80,81]. Recent population-based epidemiological studies also demonstrated that the elevation of low-carbon and low-double-bond content TG species significantly increased the risk of type 2 diabetes [82]. Our lipidemic data found that HTE supplementation exhibited anti-dyslipidemia effects by increasing the level of TG (16: 1/18: 2/20: 4), TG (16: 0/18: 3/22: 6), TG (18: 3/18: 2/22: 6), TG (18: 3/18: 3/22: 6), and TG (18: 3/20: 5/22: 6), and decreasing the level of TG (18: 0/18: 0/18: 1), TG (18: 0/18: 1/18: 1), TG (16: 0/18: 1/20: 3), TG (21: 1/18: 1/18: 1), TG (18: 0/18: 0/20: 4), and TG (18: 1/18: 1/22: 3) in the liver of *Lepr*^−/−^ rats. 

Glycerophospholipids are important cellular membrane components that regulate membrane function, including fluidity and homeostasis [83,84]. According to a previous study, defective PC biosynthesis leads to decreased VLDL synthesis and secretion, promoting NAFLD development [85]. Liu et al. reported that PCs were significantly reduced in nonalcoholic fatty livers induced by a western diet but increased significantly after highland barley β-glucan intervention [86]. Additionally, it was reported that an increase in hepatic steatosis was associated with an imbalance in the metabolism of PEs as well as the alteration in PEs content [87]. In the present study, HTE supplementation recovered the reduction of five PCs (PC (36: 2), PC (36: 4), PC (37: 2), PC (37: 5), PC (42: 7)) and 5 PEs (PE (34: 2), PE (36: 2), PE (36: 3), PE (36: 4), PE (40: 5)) in the liver of *Lepr*^−/−^ rats. PSs are multifunctional bioactive lipids and are essential to maintaining healthy nerve cell membranes and myelin [88]. CLs form the inner mitochondrial membrane, and play an important role in maintaining the stability and activity of mitochondrial proteins [89]. HTE supplementation reversed the reduction of two PSs (PS (36: 4), PS (38: 4)) and one CL (CL (20: 5/18: 2/18: 2/18: 2)) in liver of *Lepr*^−/−^ rats. In summary, our lipidemic data suggested that HTE supplement could restore lipid homeostasis by reducing glycerolipid levels and recovering glycerophospholipid levels in the liver of *Lepr*^−/−^ rats. However, the mechanisms of HTE regulating fatty-acyl-chain lipids composition of TGs class and glycerophospholipids metabolic homeostasis are needed to be investigated in the future.

Growing evidences suggest that gut microbiota dysbiosis contributes to obesity and multiple metabolic complications, and obese individuals often have a lower gut microbiota diversity than that of lean individuals [48,90,91]. In our study, along with the obesity prevention by HTE supplementation, we observed an obviously raised diversity of the gut microbiota in the KH group, which corresponds to previous research that tea could remarkably increase gut microbiota diversity [92]. It has been reported that the ratio of *Firmicutes* to *Bacteroidetes* (F/B) is positively correlated with obesity [93]. A greater abundance of *Firmicutes* facilitates energy harvesting and promotes obesity [48]. Chen et al. previously reported that Fuzhuan brick tea significantly decreased the ratio of *Firmicutes* to *Bacteroidetes*, which helped to reduce obesity in mice [94]. Our current results showed that the ratio of *Firmicutes* to *Bacteroidetes* (F/B) in *Lepr*^−/−^ rats was dramatically increased when compared to WT rats, whereas it was significantly decreased by HTE supplementation, which may also explain the improvement of HTE on obesity. To analyze the specific cluster of microbiotas with significant alteration by HTE supplementation, we found that SCFA-producing microbiota, such as unidentified_*Ruminococcaceae* and *Faecalibaculum*, increased in the KH group. It has been reported that *Ruminococcaceae* is a major butyrate producer, and its levels are dramatically reduced in patients with inflammatory bowel disease [95]. Wang et al. found that colitis was ameliorated by increasing *Ruminococcaceae* growth, which increased butyrate production [96]. Several papers reported that butyrate could prevent and treat obesity and obesity-related diseases [97,98]. In addition, a study reported that the increased abundance of *Ruminococcaceae* reduced arterial stiffness [99]. There is an association between arterial stiffness and cardiovascular diseases, while obesity is a dangerous risk factor for arterial stiffness and cardiovascular disease [100,101]. *Faecalibaculum* has been reported to have a higher fermentation capability, especially for the production of butyrate [102]. Our results showed that HTE supplementation increases the level of SCFAs in *Lepr*^−/−^ rats. SCFAs are organic fatty acids that have fewer than six carbon atoms and are considered a key mediator of the beneficial effects elicited by gut microbiota. Accumulating evidence suggests that SCFAs contribute to the improvement of physiological status of hosts by targeting multiple pathways [103,104]. Recent studies have shown that SCFAs can reduce inflammation and maintain blood glucose stability [104]. Additionally, the SCFAs also increase fatty acid oxidation inhibit fatty acid synthesis, and increase heat production to prevent obesity [105]. The results of our study indicated that the propanoic acid, butyric acid, isobutyric acid, pentanoic acid, as well as isopentanoic acid levels in the feces of the *Lepr*^−/−^ rats increased significantly after supplementation with HTE. This may explain a further mechanism by which HTE improves obesity related disorder. Additionally, the present results showed that HTE supplementation significantly increased the abundance of *Mucispirillum*. Previous published papers reported that *Mucispirillum* can antagonize the toxicity of *Salmonella* and help mice resist colitis [106]. However, HTE supplementation significantly reduced *Lactobacillus* abundance in *Lepr*^−/−^ rats. Studies reported that *Lactobacillus* may function as preventing HFD-induced obesity [107]. However, *Lactobacillus* functions are not consistent. According to Nie et al., *Lactobacillus* abundance is positively related to fasting blood glucose and HOMA-IR [108]. Sato et al. found a decline in butyrate-producing bacteria as well as an increase in *Lactobacillus* species and opportunistic pathogens in the development of type 2 diabetes [109]. Therefore, further research is required to understand the relationship between *Lactobacillus* and obesity. Altogether, the present study found that prevention of obesity complication by HTE may be possible through maintaining homeostasis of gut microbiota, increasing specific SCFA-producing bacteria, balancing hepatic lipid classes, and elevating certain hepatic lipid classes in *Lepr*^−/−^ rats.

## 5. Conclusions

In conclusion, our results showed that long term HTE supplementation remarkably reduced excessive fat accumulation, as well as ameliorated hyperlipidemia and hepatic steatosis in *Lepr*^−/−^ rats. In addition, HTE increased gut microbiota diversity and restored the relative abundance of the microbiota responsible for producing short chain fatty acids, including *Ruminococcaceae*, *Faecalibaculum*, and *Veillonellaceae*, etc. Hepatic lipidomics analysis found that HTE significantly recovered glycerolipid and glycerophospholipid classes in the liver of *Lepr*^−/−^ rats. By combination of gut microbiota and hepatic lipidomics data, our results indicated that prevention of obesity related metabolic disorder by HTE may be possible through maintaining homeostasis of gut microbiota and certain hepatic glycerolipid and glycerophospholipid classes.

## Figures and Tables

**Figure 1 foods-11-02939-f001:**
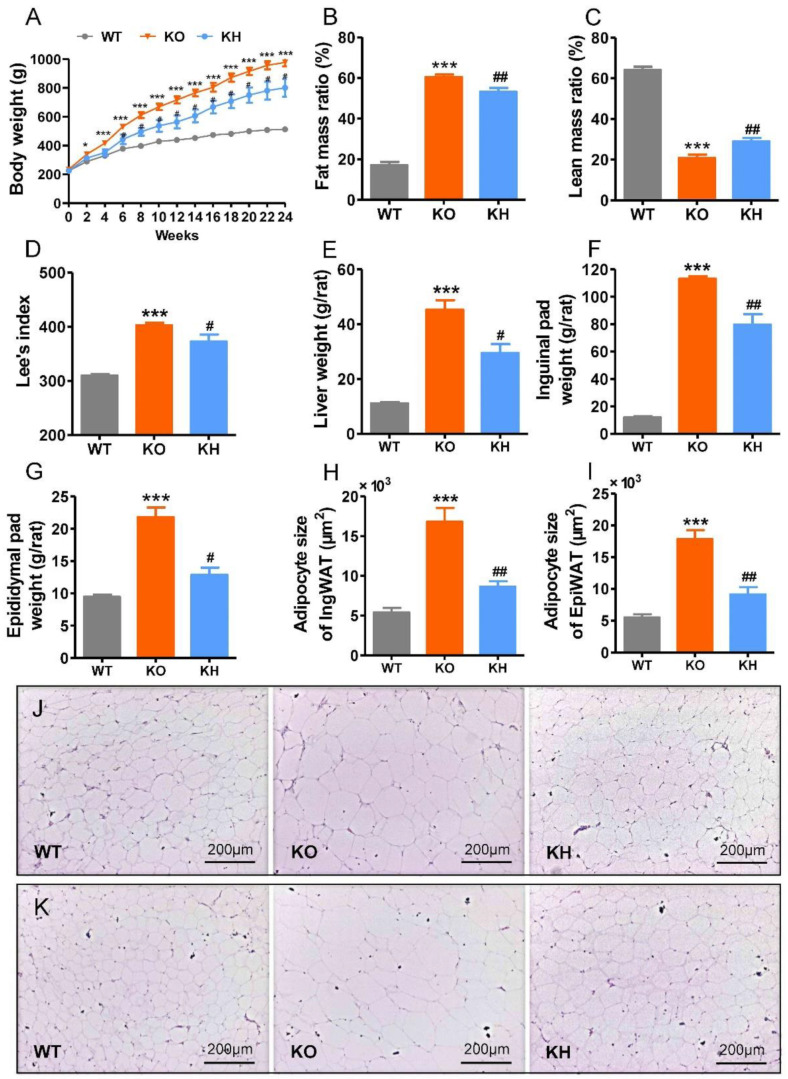
HTE attenuated phenotypes of obesity in *Lepr*^−/−^ rats. HTE, Huangshan Maofeng green tea extracts; WT, wild type littermates; KO, leptin receptor knockout rats; KH, leptin receptor knockout rats with HTE administration. (**A**) Body weight; (**B**) fat mass ratio; (**C**) lean mass ratio; (**D**) Lee’s index; (**E**) liver weight; (**F**) inguinal fat weight; (**G**) epididymal fat weight; (**H**) adipocyte size of inguinal fat; (**I**) adipocyte size of epididymal fat; (**J**,**K**) Hematoxylin–eosin staining of the inguinal fat and epididymal fat (100 × magnification). * *p* < 0.05, *** *p* < 0.001, compared to WT group; # *p* < 0.05, ## *p* < 0.01 versus KO group (*n* = 4–6, means ± SEM).

**Figure 2 foods-11-02939-f002:**
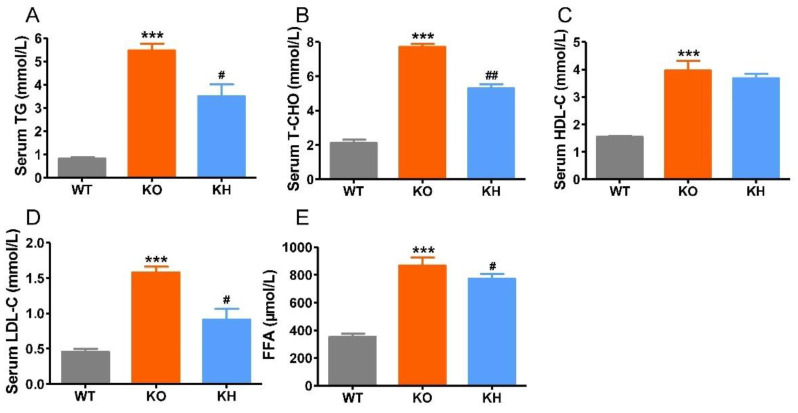
The serum lipid panel of each group of rats. (**A**) Total cholesterol; (**B**) triglyceride; (**C**) high-density lipoprotein cholesterol; (**D**) low-density lipoprotein cholesterol; (**E**) free fatty acids. *** *p* < 0.001 versus WT group; # *p* < 0.05, ## *p* < 0.01 versus KO group (*n* = 5, means ± SEM).

**Figure 3 foods-11-02939-f003:**
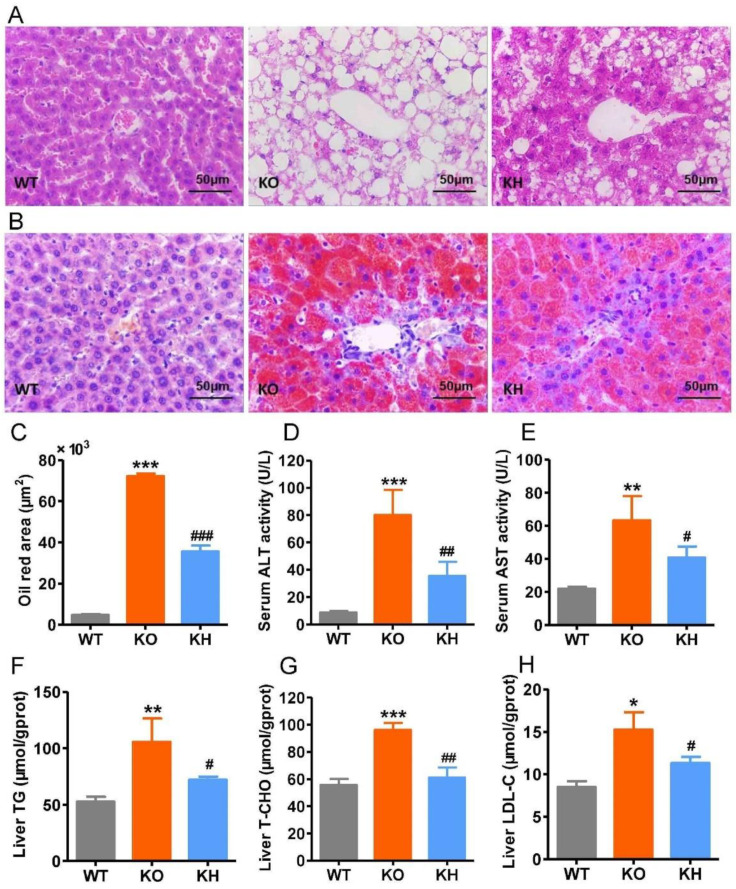
HTE ameliorated hepatic steatosis in *Lepr*^−/−^ rats. (**A**,**B**) Histological examination of liver structure with H & E staining and ORO staining; (**C**) oil red area of the liver tissue section; (**D**,**E**) the activity of serum ALT and AST, respectively; (**F**) triglyceride levels in liver; (**G**) total cholesterol levels in liver; (**H**) low-density lipoprotein cholesterol levels in liver. * *p* < 0.05, ** *p* < 0.01, *** *p* < 0.001, compared to WT group; # *p* < 0.05, ## *p* < 0.01, ### *p* < 0.001 versus KO group (*n* = 4–6, means ± SEM).

**Figure 4 foods-11-02939-f004:**
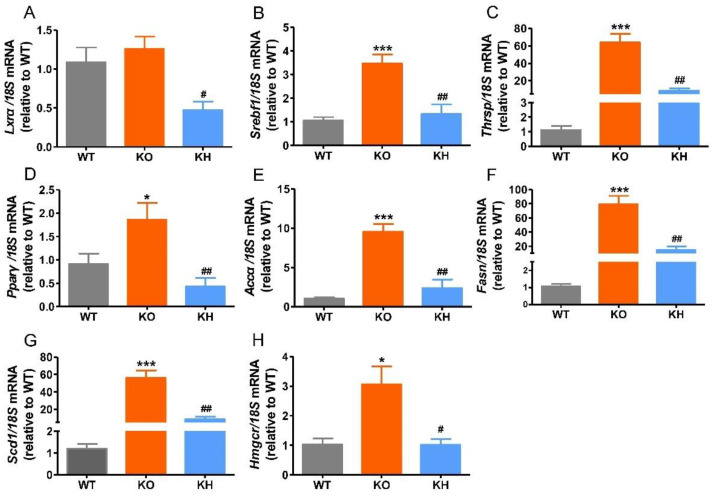
HTE reduced the expressions of hepatic lipogenesis related genes in *Lepr*^−/−^ rats. (**A**) *Lxrα*; (**B**) *Srebf1*; (**C**) *Thrsp*; (**D**) *Pparγ*; (**E**) *Accα*; (**F**) *Fasn*; (**G**) *Scd1*; (**H**) *Hmgcr*. * *p* < 0.05, *** *p* < 0.001, compared to WT group; # *p* < 0.05, ## *p* < 0.01 versus KO group (*n* = 5, means ± SEM).

**Figure 5 foods-11-02939-f005:**
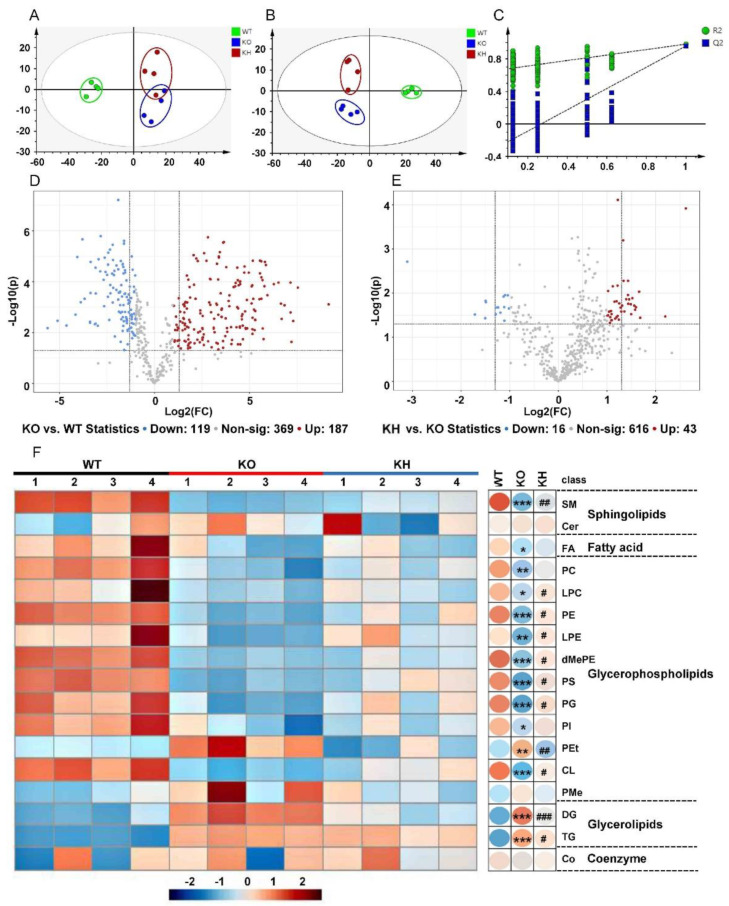
HTE altered lipid classes in liver of *Lepr*^−/−^ rats. (**A**) PCA score plots for the hepatic lipids (R^2^X = 0.907; Q^2^ = 0.518); (**B**) PLS–DA score for the hepatic lipids (R^2^X = 0.864; R^2^Y = 0.994; Q^2^ = 0.905); (**C**) PLS–DA model validation (R^2^ = 0.645; Q^2^ = −0.35); (**D**) Volcano plot displaying differential lipid classes between KO and WT; (**E**) Differential lipid classes between KH and KO. In the volcano plot, a number of significantly different lipid species were filtered out based on the criteria of FC ≥ 2 or ≤ 0.5 and *p* ≤ 0.05. Red (up) and blue (down) dots represent significant differentials of lipid species, whereas gray dots represent no significant differences of lipid species (*n* = 4). (**F**) Heat map representing lipid class levels. Values that were higher and lower than the mean will be shown in red and blue boxes, respectively. * *p* < 0.05, ** *p* < 0.01, *** *p* < 0.001, compared to WT group; # *p* < 0.05, ## *p* < 0.01, ### *p* < 0.001 versus KO group (*n* = 4, means ± SEM).

**Figure 6 foods-11-02939-f006:**
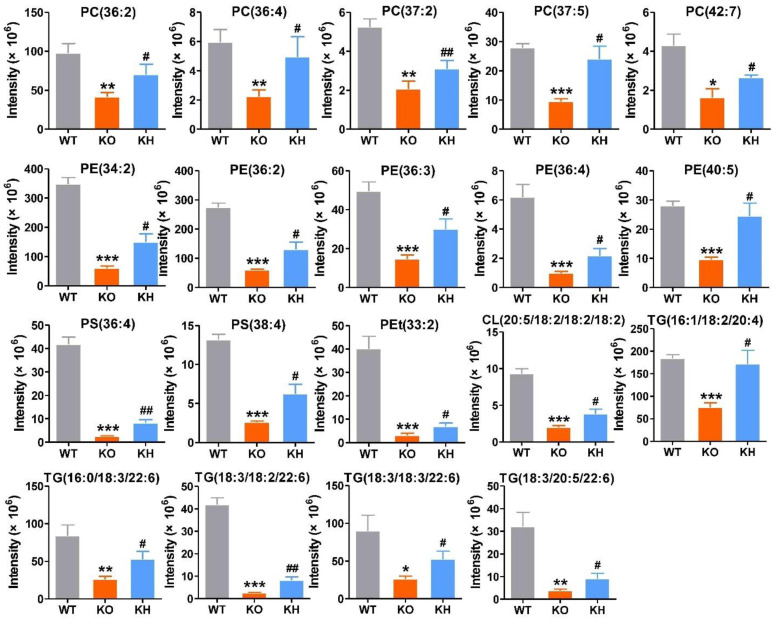
Nineteen lipid species were significant reversibly increased by HTE in liver of *Lepr*^−/−^ rats. Values are the mean ± SEM (*n* = 4). * *p* < 0.05, ** *p* < 0.01, *** *p* < 0.001, compared to WT group; # *p* < 0.05, ## *p* < 0.01 versus KO group.

**Figure 7 foods-11-02939-f007:**
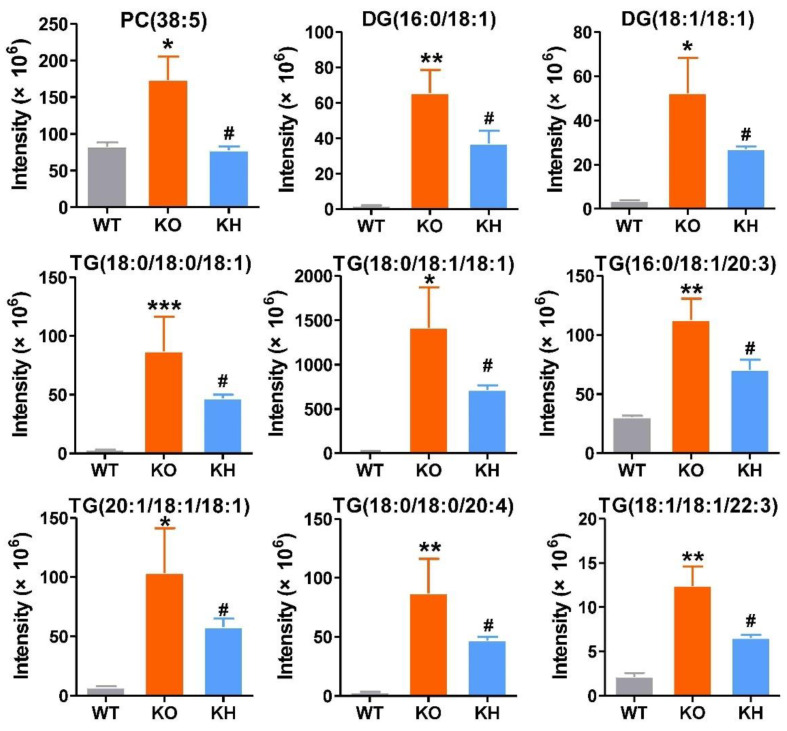
Nine lipid species were remarkable reversibly decreased by HTE in liver of *Lepr* rats. Values are the mean ± SEM (*n* = 4). * *p* < 0.05, ** *p* < 0.01, *** *p* < 0.001, compared to WT group; # *p* < 0.05 versus KO group.

**Figure 8 foods-11-02939-f008:**
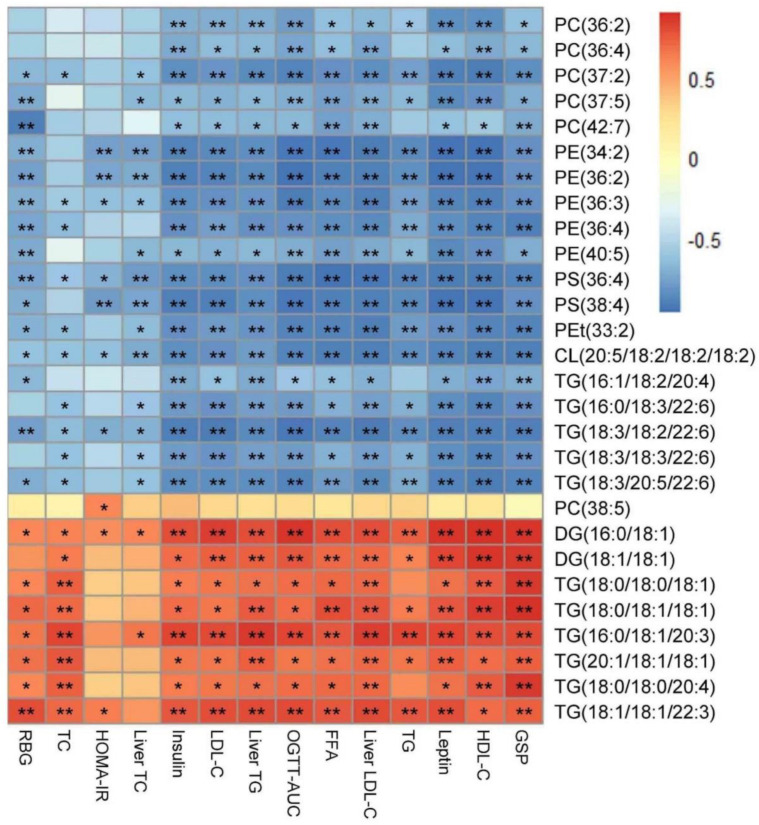
Spearman’s correlation between lipid species and obesity related indexes. Significant correlations are indicated by * *p* < 0.05, ** *p* < 0.01. RBG, random blood glucose; HOMA−IR, homeostatic model assessment of insulin resistance; OGTT-AUC, area under curve of oral glucose tolerance test; GSP, glycated serum protein level. The RBG, HOMA−IR, OGTT−AUC, GSP, insulin, and leptin levels are showed in Appendix A.

**Figure 9 foods-11-02939-f009:**
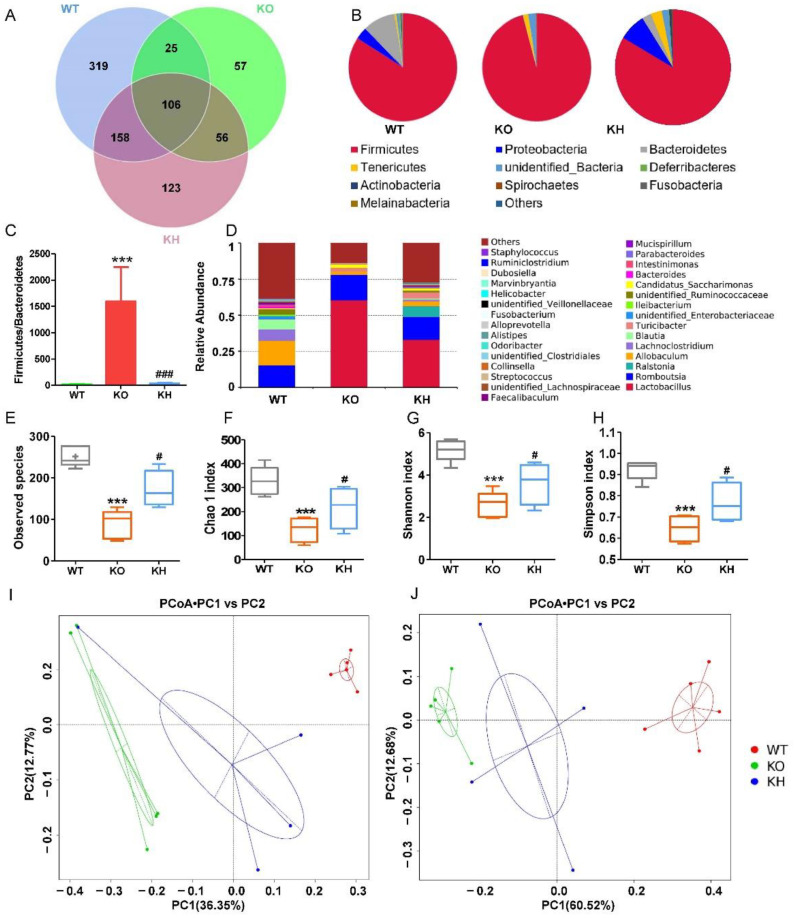
HTE increased gut microbiota diversity in *Lepr*^−/−^ rats. (**A**) Venn diagram of gut microbiota among the various groups; (**B**) the abundance of the top 10 phylum among the various groups; (**C**) the ratio of relative abundance of *Firmicutes* to *Bacteroidetes*; (**D**) the abundance of the top 30 genus among the various groups; (**E**–**H**) assessment of the alpha diversity of gut microbiota based on different indices; (**I**,**J**) principal coordinates analysis of gut microbiota based on unweighted UniFrac and weighted UniFrac. *** *p* < 0.001 versus WT group; # *p* < 0.05, ### *p* < 0.001, versus KO group (*n* = 4–5, means ± SEM).

**Figure 10 foods-11-02939-f010:**
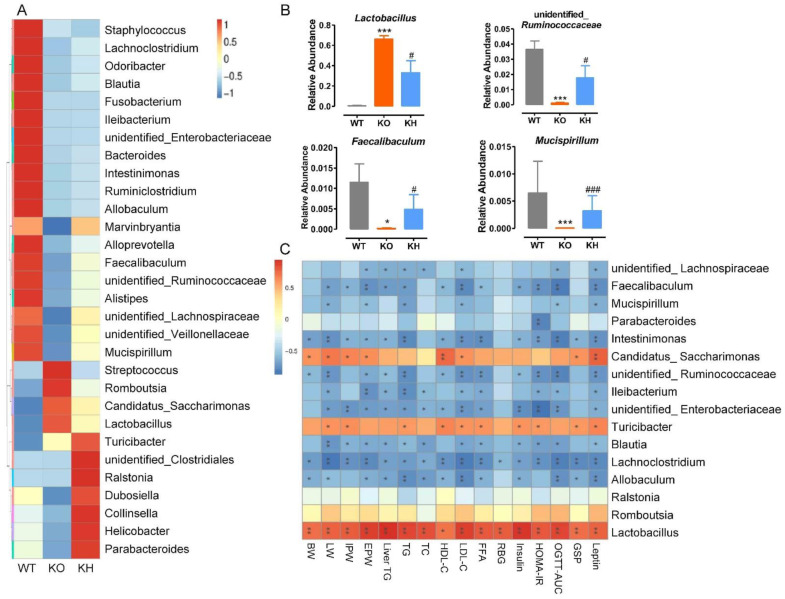
HTE altered the gut microbiota composition in *Lepr*^−/−^ rats. (**A**) The heatmap of the top 30 genera among the various groups; (**B**) relative abundance of the gut microbiota at the genus level (*p* < 0.05); (**C**) heatmap of spearman’s correlation between microbiota and obesity related parameters, significant correlations are marked as * *p* < 0.05, ** *p* < 0.01. BW, body weight of rats; LW, liver weight; IPW, inguinal fat weight; EPW, epididymal fat weight. * *p* < 0.05, *** *p* < 0.001 versus WT group; # *p* < 0.05, ### *p* < 0.001 versus KO group (*n* = 4–5, means ± SEM).

**Table 1 foods-11-02939-t001:** Contents of catechins, polysaccharide, caffeine, and amino acid in lyophilized powder of tea water extracts.

Compound	LTE (mg/g)	HTE (mg/g)
EGCG	127.77 ± 8.06	178.14 ± 2.37 **
EC	50.49 ± 5.60	63.54 ± 2.37
GCG	63.33 ± 0.93	54.52 ± 0.50 **
GC	44.53 ± 0.49	42.62 ± 0.36 *
C	27.71 ± 0.46	43.02 ± 2.34 **
Total catechins	313.83 ± 14.45	381.84 ± 4.56 *
Polysaccharide	188.02 ± 4.33	153.89 ± 1.36 ***
Caffeine	111.25 ± 0.52	106.92 ± 0.93 *
Theanine	20.74 ± 0.14	30.51 ± 1.43 ***
Total amino acids	50.31 ± 0.42	74.63 ± 5.58 ***

HTE, Huangshan Maofeng green tea extracts; LTE, large yellow tea extracts. Total catechins are sum amount of EGCG, EC, GCG, GC, and C; total amino acids are amount of theanine plus other free amino acids. * *p* < 0.05; ** *p* < 0.01; *** *p* < 0.001 when compared with LTE (*n* = 3, means ± SEM).

## Data Availability

The data are available from the corresponding author.

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
