# Peer review of "Huangshan Maofeng Green Tea Extracts Prevent Obesity-Associated Metabolic Disorders by Maintaining Homeostasis of Gut Microbiota and Hepatic Lipid Classes in Leptin Receptor Knockout Rats"

_foods, 2022, doi:10.3390/foods11192939_

Round 1
Reviewer 1 Report
In the research article by Wu and colleagues, the authors investigated the effect of green tea extract on metabolic disorders linked with obesity in Leptin receptor knockout rats. They observed reduced hyperlipidemia and steatosis in the experimental cohorts. Along with the histopathological evaluations, serum levels of triglyceride, cholesterol, and other biochemical markers of liver functions were also measured. Although this is a well-planned and well-written manuscript, some minor issues need to be addressed.
1. In table 1, the authors reported the active component in each fraction, either HTE or LTE. The total weight of all these components is greater (1128.62 mg) than a gram (1000 mg) in HTE. Please check the calculation and yield.
2. When were rats administered HTE extract? Intervention and prevention term in the same experiment is confusing. Provide detailed experimental design and protocol.
Reviewer 2 Report
Comments to the Authors
The manuscript from Guohuo Wu et al. entitled “Huangshan Maofeng Green Tea Extracts Prevent Obesity-Associated Metabolic Disorders by Maintaining Homeostasis of Gut Microbiota and Hepatic Lipid Classes in Leptin Receptor Knockout Rats” seems interesting. There are points which need to be addressed.
1. Abstract, several words should be fully spelled (TG, DG, PC, PE, and TG).
2. Lines 110-, how was HTE given orally? If by gavage, what was a solvent for HTE and how was the concentration?
3. Line 120, which adipose tissues were excised?
4. Lines 143-, for using the ΔΔCt method, preliminary experiments are necessary to check the PCR amplification efficiency for each gene. Did authors check it for all the primers?
5. Line 175, SCFA should be fully spelled in first appearance.
6. Lines 183-, authors describe ANOVA. Did Authors perform a test for normality?
7. Table 1, in this study authors focused on HTE; therefore, LTE should be a control, HTE can be right side, and asterisks can be on the numbers of HTE.
8. Figure 1B, how were fat and lean mass measured?
9. Figures 1E-G, because body weights were significantly different, how were the weights of liver and fats relative to whole body weight? Authors can show them.
10. Figure legends, the means of WT, KO, and KH should be described.
11. Figure 2C, how do authors evaluate the alteration of HDL levels? It should be discussed in Lines 456-.
12. Lines 245-, WT group exhibited larger polygonal cells with clear round nuclei and a reduction in lipid accumulation. Because WT was a control, an expression “a reduction” seems inappropriate.
13. Line 247, what is amyloidosis? How was it observed?
14. Lines 250- and Figure 3, what are ALT and AST activities? Do they mean enzyme activities? This appears contradictory to the aforementioned in M&M.
15. Figure 3, the pictures of histopathology of the livers are inappropriate. They need to be in lower magnification with central vein and portal vein, ideally.
16. Figure 3, what does the unit mean in the longitudinal axis “gprot?”
17. Line 286, what is PSL-DA?
18. Figure 8, the legend is not proper, including the spelling of HOMA-IR.
19. Line 426, full spell of HMGT already appeared in Introduction.
20. Line 528, there are two “our.”
21. Lines 550 and 559, abbreviations are not necessary for IBD and FBG, because they appear there only.
22. Lines 545-559, authors described butyrate, IBD, arterial stiffness, and “multiple pathways.” Authors can describe the relations of obesity and its related disorders with them.
23. Lines 561-, This may explain a further mechanism by which HTE improve obesity related disorder. Readers may not understand the mechanism how HTE improved obesity related disorder. Authors need to describe in detail.
24. Line 578, does “hyperlipidemia” include HDL?
25. Leptin controls appetite. Authors should describe data of food intake.
Reviewer 3 Report
The study entitled “Huangshan Maofeng Green Tea Extracts Prevent Obesity-Associated Metabolic Disorders by Maintaining Homeostasis of Gut Microbiota and Hepatic Lipid Classes in Leptin Receptor Knockout Rats by Wu et al., is well designed and performed, and the results are clearly presented in the manuscript. The authors demonstrated that Huangshan Maofeng green tea water extract (THE) might prevent obesity-related metabolic disorders by maintaining the homeostasis of gut microbiota and certain hepatic glycerolipid and glycerophospholipid classes.
Few sentences are required language correction, grammar, and spell-check.
Round 2
Reviewer 2 Report
According to the comments from the Reviewers, the Authors responded adequately and conducted several modifications appropriately. This seems a quite well-written and reshaped manuscript. Therefore, this can be suitable for publication in the journal.